# Comparison of UAS and Sentinel-2 Multispectral Imagery for Water Quality Monitoring: A Case Study for Acid Mine Drainage Affected Areas (SW Spain)

**Melisa A. Isgró** [1,2], **M. Dolores Basallote** [1,*], **Isabel Caballero** [3] and **Luis Barbero** [2]

1 Research Center on Natural Resources, Health, and the Environment, University of Huelva, 21071 Huelva, Spain

2 Department of Earth Sciences, University of Cádiz, Av. República Saharaui s/n, Puerto Real, 11510 Cádiz, Spain

3 Instituto de Ciencias Marinas de Andalucía (ICMAN-CSIC), Campus Universitario Río San Pedro, s/n, Puerto Real, 11510 Cádiz, Spain

* Correspondence: maria.basallote@dct.uhu.es; Tel.: +34-959-219-835

**Abstract:** Uncrewed Aerial Systems (UAS) and satellites are used for monitoring and assessing the quality of surface waters. Combining both sensors in a joint tool may scale local water quality retrieval models to regional and global scales by translating UAS-based models to satellite imagery. The main objective of this study is to examine whether Sentinel-2 (S2) data can complement UAS data, specifically from the MicaSense RedEdge MX-Dual sensor, for inland water quality monitoring in mining environments affected by acid mine drainage (AMD). For this purpose, a comparison between UAS reflectance maps and atmospherically corrected S2 imagery was performed. S2 data were processed with Case 2 Regional Coast Colour (C2RCC) and Case 2 Regional Coast Colour for Complex waters (C2X) atmospheric correction (AC) processors. The correlation between the UAS data and the atmospherically corrected S2 data was evaluated on a band-by-band and a pixel-by-pixel basis, and the compatibility of the spectral data was analyzed through statistical methods. The results showed C2RCC and C2X performed better for acidic greenish-blue and non-acidic greenish-brown water bodies concerning the UAS data than for acidic dark reddish-brown waters. However, significant differences in reflectance between the UAS sensor and both S2 AC processors have been detected. The poor agreement between sensors should be considered when combining data from both instruments since these could have further consequences in developing multi-scale models.

**Keywords:** abandoned mine; acidic water; surface water monitoring; drone; Copernicus programme

## 1. Introduction

Remote sensing techniques have gained popularity in recent years and the improvement in terms of spectral, spatial, and temporal resolutions of the sensors have increased their use for analysis and monitoring in different fields of study [1–3]. Lately, Uncrewed Aerial Systems (UAS) have grown as an innovative platform for acquiring ultra-high-resolution images at low altitudes, providing both high spatial centimeter-scale and flexible temporal resolution, at an increasingly affordable price. At the same time, recently launched satellites withing the Copernicus programme, such as Sentinel-2A/B (S2), can also provide a medium spatial resolution (10–20–60 m) with a revisit frequency of five days (at the equator) but delivering global coverage. Each of these data platforms is generally used separately even though it is known that they can be complementary and have strong synergies [4]. While the main applications that exploit the UAS/Satellite complementarities are precision agriculture and ecology [5–8], water resources in inland waters remain an underexplored field [9].

UAS imagery can be used to complement satellite imagery with cloud cover [10] and increase spatial resolution. These characteristics are especially relevant to mapping

small water bodies, where the satellite resolution does not allow the recognition of events occurring at a local scale [11–13] or in the case of highly dynamic scenarios such as coastal areas [14], and to avoid mixed pixels particularly near the water boundaries [15]. On the other hand, the use of satellite data can scale local models to regional and global scales by translating UAS-based models to satellite imagery [4,7]. This would be impossible using only UAS since they have only a few km$^2$ swaths, due to the short flight time and the national legislation regarding air traffic and people's privacy. Nevertheless, the differences in acquisition time, resolution, environmental conditions, spectral bands, sensor position, and atmospheric effects (among others factors), may result in discrepancies in the performances of the sensors [4]. Thus, it is imperative to study the consistency between UAS and satellite observations, which may affect the multi-scale model proficiency.

This study focuses on the comparison of reflectance data acquired by the commercial sensor Micasense RedEdge-MX Dual onboard an UAS (DJI Matrice 210 V2) and by S2 in small water bodies affected by acid mine drainage (AMD) at different extents. AMD is the formation and movement of highly acidic water rich in metals and sulfate due to the oxidation of sulfide-bearing minerals exposed to atmospheric weathering processes, which frequently occurs in metal mine (and coal mine) areas. The result is a highly toxic solution that causes harmful effects affecting animals, plants, and even humans [16–18]. Data acquisition was carried out in the Tharsis mine complex, at the Iberian Pyrite Belt (IPB) (SW Spain; Figure 1), where the historical mining activity has left a legacy of several abandoned sites, which became the sources of AMD in the area. The acid leachates have reached the aquifers and river streams by percolation and runoff, being responsible for the degradation of the water system and the formation of water bodies containing highly acidic and metal-polluted water [16,18,19]. This research will help with understanding the coupling between chemical and optical properties of water and how remote sensing is applied to resolving these AMD-affected waters cases. AMD-waters have a complex composition due to their physicochemical characteristics, which are also observed in their different colors. Ocean Colour remote sensing usually distinguish two water classes: Case 1 waters which are ocean waters whose optical properties are determined by phytoplankton and other covarying compounds and Case 2 waters which are coastal and inland waters whose optical properties are determined by other constituents, such as mineral particles and Chromophoric Dissolved Organic Matter (CDOM), and their concentrations do not covary with phytoplankton [20]. Few studies have focused on quantifying water quality parameters in mining areas and most of them have been based on satellite datasets [21–26], whereas the application of UAS is rather recent [27]. The novelty of this study consists in exploring the possibility of combining different scale sensors in order to achieve multi-scale models. Therefore, the main objective is to examine whether S2 data can complement UAS data, specifically from the MicaSense RedEdge MX-Dual sensor, for inland water quality monitoring in mining environments. For this purpose, a comparison between UAS reflectance maps and atmospherically corrected S2 imagery was performed.

In addition, S2 requires atmospheric correction (AC) to subtract the atmospheric and sunglint contribution from the top of the atmosphere (TOA) signal, and there are several AC processors available for inland and coastal waters [28]. In other water masses, these models have been satisfyingly applied to process from TOA to BOA (bottom of atmosphere) reflectance, a step required for monitoring terrestrial, emerged intertidal, benthic or coastal ecosystems [2,28–30]. Hence, for this study, two AC processors were selected to be applied on S2 data: Case 2 Regional Coast Colour (C2RCC) and Case 2 Regional Coast Colour for Complex waters (C2X).

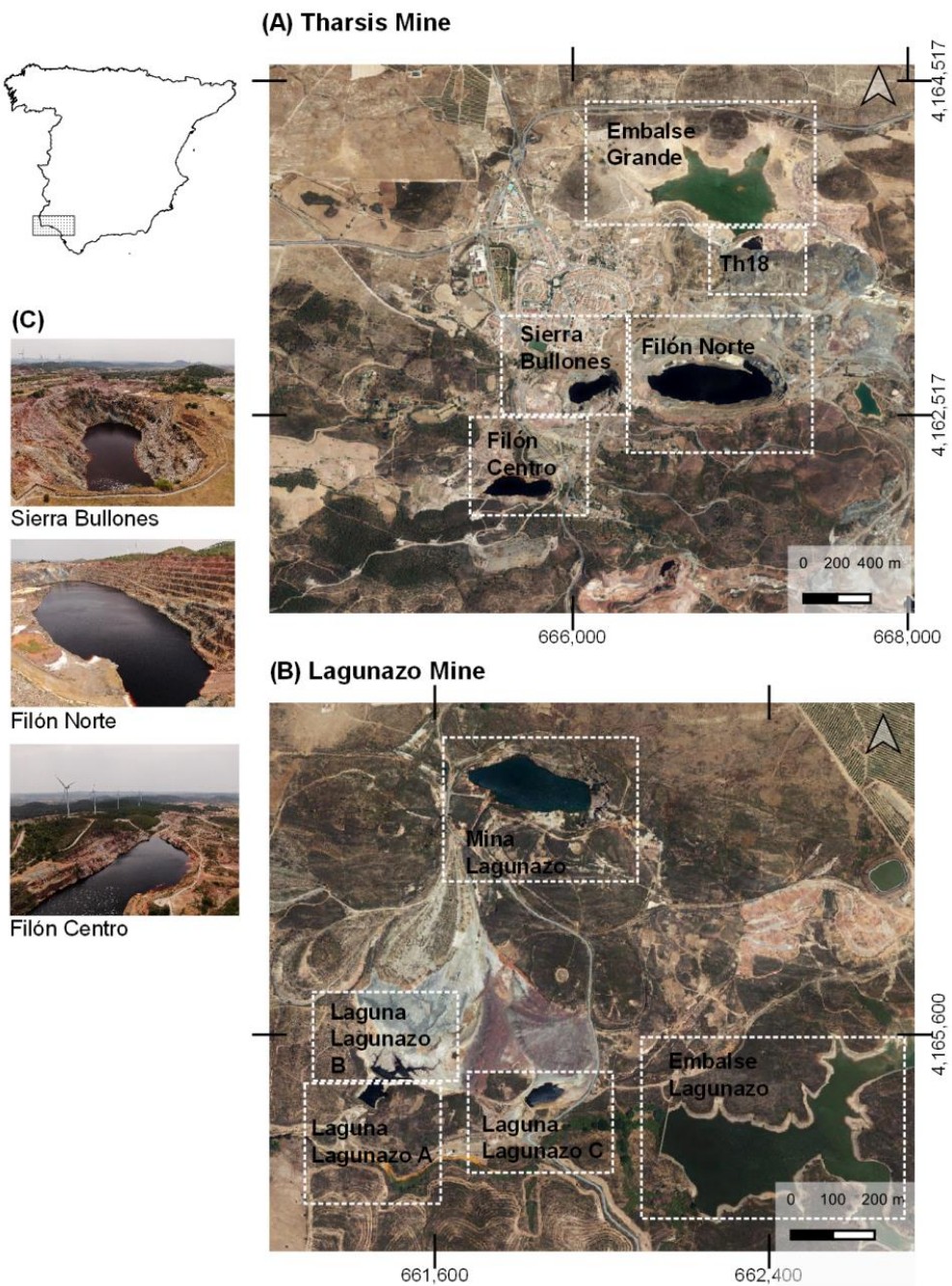

**Figure 1.** Location map of sampling sites. Five water bodies (Filón Norte (FN), Filón Centro (FC), Sierra Bullones (SB), Embalse Grande (EG), and Th18) sampled in the Tharsis Mine (**A**), and five water bodies (Mina Lagunazo (ML), Embalse Lagunazo (EL), Laguna Lagunazo A, B, and C (LLA, LLB, and LLC, respectively)] in Lagunazo Mine site (**B**). (**C**) corresponds to aerial images of different inland water masses evaluated in this study.

## 2. Materials and Methods

### 2.1. Study Area and Water Physicochemical Parameters

The study area was the IPB (SW Spain), which contains one of the greatest concentrations of polymetallic massive sulfide mineralization on Earth, with more than 1700 Mt of sulfide ore estimated (mined and reserves) [17,31]. As a result of the historical mineral exploitation, there are numerous abandoned mining sites, such as the Tharsis and Lagunazo mines [32], which were the areas sampled for this study.

Figure 1 shows the water bodies selected for this study: Embalse Grande (EG), Sierra Bullones (SB), Filón Centro (FC), Filón Norte (FN), and TH18 from Tharsis mine (Figure 1A); and Mina Lagunazo (ML), Embalse Lagunazo (EL), Laguna Lagunazo A, B and C (LLA, LLB, LLC) from Lagunazo mine site (Figure 1B). EG and EL are relatively clean water reservoirs used for agricultural purposes. SB, FC, FN, and ML are pit lakes flooded with acidic waters, and TH18, LLA, LLB, and LLC are surficial AMD sites.

The study area covers a wide gradient of water compositions, from neutral to extremely acidic pH (0.01–7.76) and from low to high metal-enriched solutions (e.g., 0.05–60, 895 mg $L^{-1}$ Fe). The water composition is also reflected in the watercolor: EG and ML are nonacidic water bodies and have a greenish-brown color, ML is acidic and presents greenish-blue color and the rest of them are dark reddish-brown acid waters with high iron concentrations. In situ physicochemical data were collected simultaneously with the UAS data acquisition. Detailed information about sampling can be found in [33]. Briefly, pH, electrical conductivity (EC), oxidation–reduction potential (ORP), and temperature (T) were measured at each sampling point with a CrisonMM40 þ multimeter, previously calibrated with certified solutions. Turbidity was measured using a Hanna HI-93703 portable turbidimeter at the water surface. Water samples were collected in HDPE bottles previously acid washed, filtered (0.45 mm pore size cellulose nitrate membrane), and $HNO_3$-acidified to pH < 2. Major elements (Al, Ba, Ca, Cu, Fe, K, Mg, Mn, Na, P, S, Si, Sr, and Zn) were determined by inductively coupled plasma-atomic emission spectroscopy (ICP-AES; PerkinElmer® Optima 3200 RL, Waltham, MA, USA) at the Institute of Environment Assessment and Water Research (IDAEA-CSIC, Barcelona, Spain).

### 2.2. Remotely Sensed Data

Two different multispectral sensors were used for this study: an airborne sensor (MicaSense RedEdge-MX Dual onboard a DJI Matrice 210 V2) and a spaceborne sensor (S2). The spatial resolution and bandwidths of the sensors used in this study are described in Table 1. Only the bands used for this study are shown in the table.

**Table 1.** Central wavelengths (nm) and bandwidths of the evaluated sensors bands.

| Sensors | Coastal Aerosol | Blue | Green | Red | Red-Edge1 | Red-Edge2 | NIR |
|---|---|---|---|---|---|---|---|
| **Sentinel-2** | | | | | | | |
| **Central Wavelength** | 443 | 490 | 560 | 665 | 704 | 740 | 865 |
| **Bandwith** | 20 | 60 | 36 | 30 | 15 | 15 | 21 |
| **MicaSense** | | | | | | | |
| **Central Wavelength** | 444 | 475 | 560 | 668 | 705 | 740 | 840 |
| **Bandwith** | 28 | 20 | 20 | 10 | 10 | 18 | 40 |

### 2.2.1. UAS Data Acquisition and Processing

The UAS DJI Matrice 210 V2, a rotary-wing quadcopter platform with vertical take-off and landing (VTOL), was used to collect aerial images at a constant flight altitude over water bodies. The sensor mounted on the UAS to acquire the imagery was a lightweight multispectral camera: MicaSense RedEdge-MX Dual, which has ten different channels. The sensor resolution is 1.2-megapixel for each of the multispectral bands and 3.6-megapixel for the RGB captures. The lens achieves a ground sampling distance (GSD) of 8 cm/pixel at 120 m above ground level (AGL). Autonomous aerial flights were performed using the DJI GS Pro planning software from 120 m AGL altitude with 80% frontal and 75% side image overlap, the grid was simple, and the speed was set at 10 m/s. It is assumed that all the multispectral imagery is in the nadir position due to the location of the camera. No Ground Control Points (GCPs) were added to the flights since the Micasesnse sensor has an integrated GPS that geo-tag each of the images acquired by the UAS. Furthermore, it was also equipped with a Downwelling Light Sensor (DLS 2), and a MicaSense's Calibrated

Reflectance Panel (CRP) to perform the radiometric calibration on the ambient light changes during the flight.

The DLS 2 and CRP are necessary for the radiometric calibration on the ambient light changes during the flight. For this purpose, a picture of the CRP was taken before and after the flight to capture the lighting conditions of the flight ensuring no shadows were covering the panel. The use of the CRP allows comparing the data acquisition over different dates or at different times of day, which is essential for a multiple flight analysis. On the other hand, the DLS 2 is located at the top of the UAS looking upwards towards the sky and measures the variations in light intensity throughout the flight by using 10 light sensors located in different planes to define the sun position and its effect on the images. This light information is stored in each TIFF image metadata and is used to correct the illumination variability during the mosaicking process [34]. By using both the DLS2 and CRP during the data collection, it is possible to choose the best calibration option to get a high-quality reflectance map during the image processing.

Pix4D mapper (Pix4D S.A., Lucerne, Switzerland) was the Structure from Motion (S*f*M) software used to process the multispectral images. Before starting the pre-processing, images were inspected and discarded if blurred. From all images captured by the UAS platform, the software creates orthomosaics of the surface reflectance values by image alignment, creation of a 3D point cloud and triangle mesh, and digital surface model (DSM), orthomosaic, and index generation (Figure S1). During these steps, the radiometric processing and calibration are performed, and different calibration methods can be applied depending on the input data and the light conditions. In this study, two methods were implemented: Camera and Sun Irradiance and Camera only [35]. For Camera and Sun Irradiance calibration, Pix4D uses the parameters written in the EXIF metadata and relate to the camera, as well as the CRP to calculate the absolute irradiance and the DLS data for normalizing each image for changes in the incoming radiation during the flight. For Camera only calibration, Pix4D also uses the parameters written in the EXIF metadata and relate to the camera and the CRP, but the DLS data is not considered. The last method is more appropriate when the sky is clear and there is no variation in the light conditions [36]. As a result of this process, ten single orthomosaics of the surface reflectance values with a GSD of 0.08 m were generated for all the flights. These reflectance maps contain reflectance values ranging between 0.0 and 1.0 for each pixel. The atmospheric correction was not considered for the UAS imagery, since it is a close-range remote sensing approach and the atmosphere layer between the UAS and the ground is so thin that it can be omitted [9,37]. Considering that the UAS data are a low altitude remote sensing tool with ultra-high spatial resolution that is not affected by the atmosphere and that field spectro-radiometric measurements were not performed, the UAS data were considered as the ground truth reference source for the remote sensing.

The UAS data acquisition was carried out on 7 October 2020 and the overpassing of S2 was the next day on 8 October 2020.

### 2.2.2. Sentinel-2 Data Acquisition and Processing

S2 Level-1C product was acquired from the ESA Copernicus Open Access Hub (https://scihub.copernicus.eu/, accessed on 1 March 2022). The study region is covered by one S2 tile (29SQB), which is a 100 km by 100 km squared orthoimage in UTM/WGS84 projection. The tile was acquired as close as possible to the date on which the UAS-based survey was carried out. Images with cloud cover and sun glint over the water bodies were discarded. S2 images were available on 8 October 2020, at 11:08:49 UTC.

Satellite imagery needs to be atmospherically corrected since the radiance reaching the sensor interacts with the atmosphere and is affected by several parameters, such as wind speed, sun glint, angle of the sun, aerosols, water vapor, and land adjacency effects, among others. AC processors can correct the "noise" from the radiance to get the water-leaving reflectance true value. Many AC processors have been tested and validated for coastal waters, but only a few studies have been performed over inland waters [20,28,38–40].

In this study, the images were processed to Level-2A (L2A) with the Case 2 Regional CoastColour processor (C2RCC) and Case 2 Regional Coast Colour for Complex waters (C2X) [40]. C2RCC is an AC processor based on artificial neural networks (ANN) that relate TOA water reflectance to radiative transfer simulations of water-leaving radiances. It uses a radiative transfer model for the atmospheric characteristics and a bio-optical model for the water characteristics. These water characteristics are parameterized considering the water reflectance in the blue spectrum (443 nm), based on bio-optical databases. The ANN are then trained based on TOA reflectance, these modelled water characteristics and environmental variables [41]. C2RCC was complemented with the CoastColour dataset to extend the range for coastal waters including extreme cases resulting in the C2X. C2X uses a special set of networks trained for extreme ranges of absorption and scattering, that is, for very turbid waters [42]. Previous studies with S2 imagery in inland waters have obtained good performance for both AC processors [19,43–46], even though for coastal waters ACOLITE has been proved to be more consistent [23]. The AC was performed through the ESA's Sentinel toolbox in the Sentinel Application Platform (SNAP).

All the bands were previously resampled to 10 m in SNAP using the bilinear upsampling method. The satellite imagery was then reprojected to WGS 84/UTM zone 29N with nearest neighbor interpolation to match the UAS data.

### 2.3. Spectral Band Comparison

The UAS data (8 cm) was resampled and aligned in the open-source geographic information system software QGIS to match the satellite pixel size of 10 m employing the bilinear interpolation method. To compare the results of both sensors datasets, for each water body the average of six $3 \times 3$ boxes were extracted from the UAS and S2. The comparison was based on pure pixels, by choosing central pixels of the water body when possible, meaning that pixels with only water cover were selected. The UAS bands were transformed into remote sensing reflectance (Rrs) with unit 1/sr by the following equation [47,48]:

$$Rrs = Rhow/\pi$$

where $\pi$ has units of steradian and Rhow is the non-dimensional water-leaving reflectance. Then the UAS-imagery [47,48] were paired with the most contemporaneous available S2 image (8 October 2020) (see detailed information on the flowchart in Figure 2).

The correlation between the UAS data and the atmospherically corrected S2 data (C2RCC and C2X) was evaluated on a band-by-band basis to determine the strength of the relationship between both sensors. For this purpose, a linear model was fitted to the UAS-S2 image pair and the coefficient of determination ($R^2$) was calculated and compared. Moreover, a pixel-by-pixel and band-by-band comparison was performed as a graphical output of the analyzed reflectance values. To validate the consistency of both sensors, the band-by-band root mean square error (RMSE), mean absolute error (MAE), and bias between all the UAS resampled pixels and the corresponding S2 data were calculated. MAE and bias provide more robust output with non-Gaussian distributions and outliers [44].

Afterwards, the compatibility of the spectral data from the UAS bands and the S2 bands was analyzed through statistical methods. The non-parametric paired-sample Wilcoxon signed-rank test was applied to compare the mean difference of reflectance values from UAS data with the S2 data band-by-band. This test is an alternative to paired t-test since the data have no normal distribution according to the results obtained by the Shapiro-Wilk test. The test will indicate whether there is a statistical difference between the two means of UAS and S2 data in order to discern if the data combination of both sensors should be successful and can be complementary. In contrast, if significant differences exist, this should be taken into account when combining data from both sensors since these differences could affect the performance of models, leading to biases in the estimation of variables, such as water quality parameters.

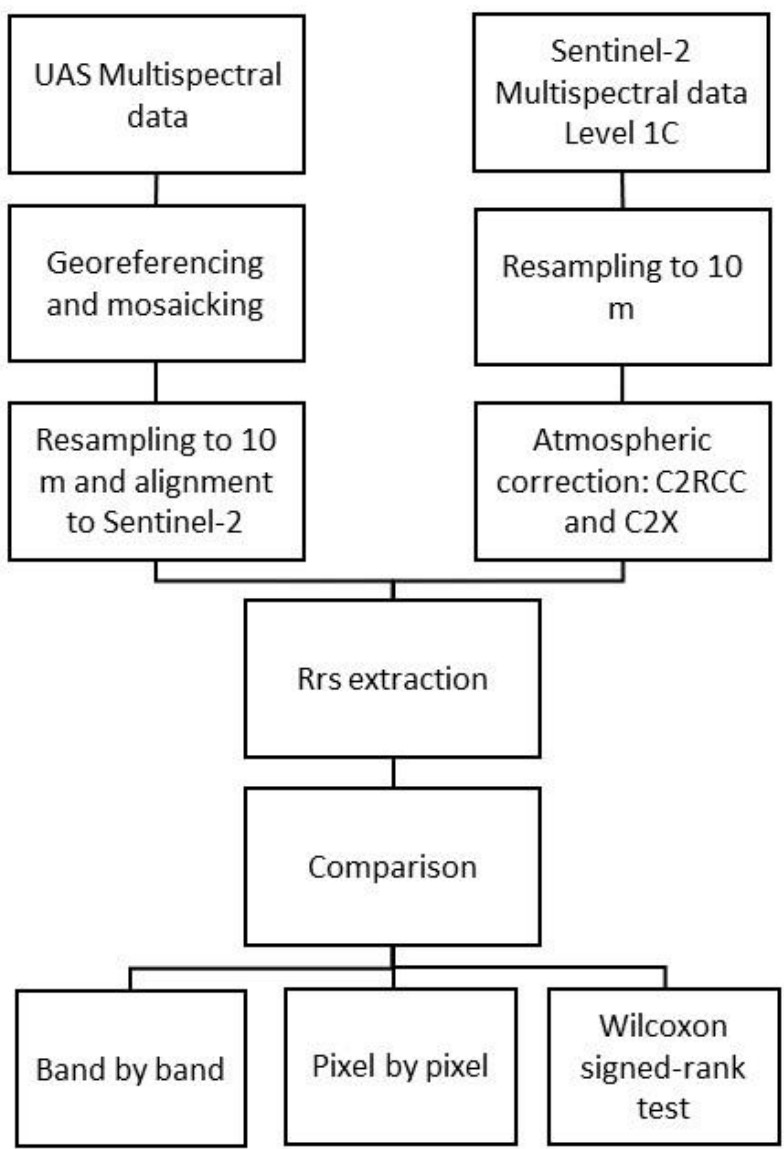

**Figure 2.** Flowchart of the methodological procedures for sensors comparison.

### 3. Results

*3.1. Reflectance Spectra*

The UAS and S2 sensors' reflectance spectra for each water body are shown in Figure 3. The S2 reflectances are atmospherically corrected with C2RCC and C2X processors. Based on the general shape of the spectra, greenish-blue and greenish-brown water bodies (ML, EG, and EL) showed a better agreement between sensors (Figure 3A,B,F). For dark reddish-brown waters (LLA, LLB, LLC, Th18, FC, FN, SB), the shape of the spectra seems to vary depending on the sensor and atmospheric correction procedure. The differences between UAS and S2 measurements are pronounced at the coastal blue and NIR bands, while reflectances are more consistent in the blue, green, red, red-edge 1, and red-edge 2 bands. Moreover, dark reddish-brown water reflectance is much lower than greenish-blue and greenish-brown waters due to the higher absorption and smaller backscattering of its water surfaces.

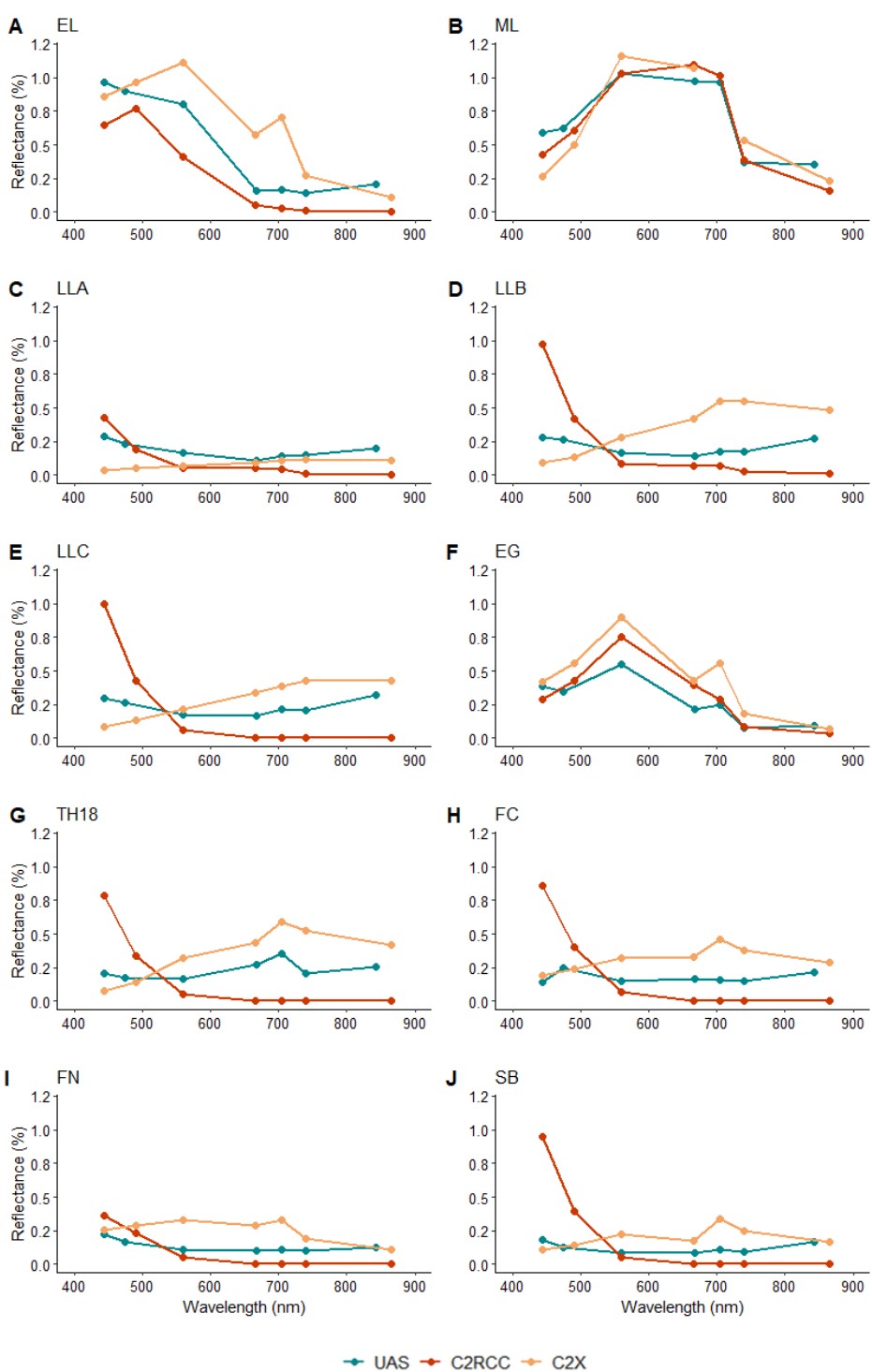

**Figure 3.** Reflectance spectra collected from Embalse Lagunazo (**A**), Mina Lagunazo (**B**), Laguna Lagunazo A (**C**), Laguna Lagunazo B (**D**), Laguna Lagunazo C (**E**), Embalse Grande (**F**), TH18 (**G**), Filón Centro (**H**), Filón Norte (**I**), and Sierra Bullones (**J**) in October with UAS and Sentinel-2 bands obtained with C2RCC and C2X atmospheric correction processing.

*3.2. Spectral Band Comparison*

The band-by-band and pixel-by-pixel comparison showed that the UAS and C2RCC data follow the same trend for the green, red, red-edge1, red-edge2, and NIR bands (Figure 4C–G). Nevertheless, C2X showed a high variability of the pixel values in all the analyzed bands. The UAS surface reflectance data has low variability in the reflectance

values of all the analyzed bands. The data showing higher reflectance in the coastal blue, blue, and green band correspond to EL, ML, and EG (Figure 4A–C) which are the greenish-blue and greenish-brown water bodies.

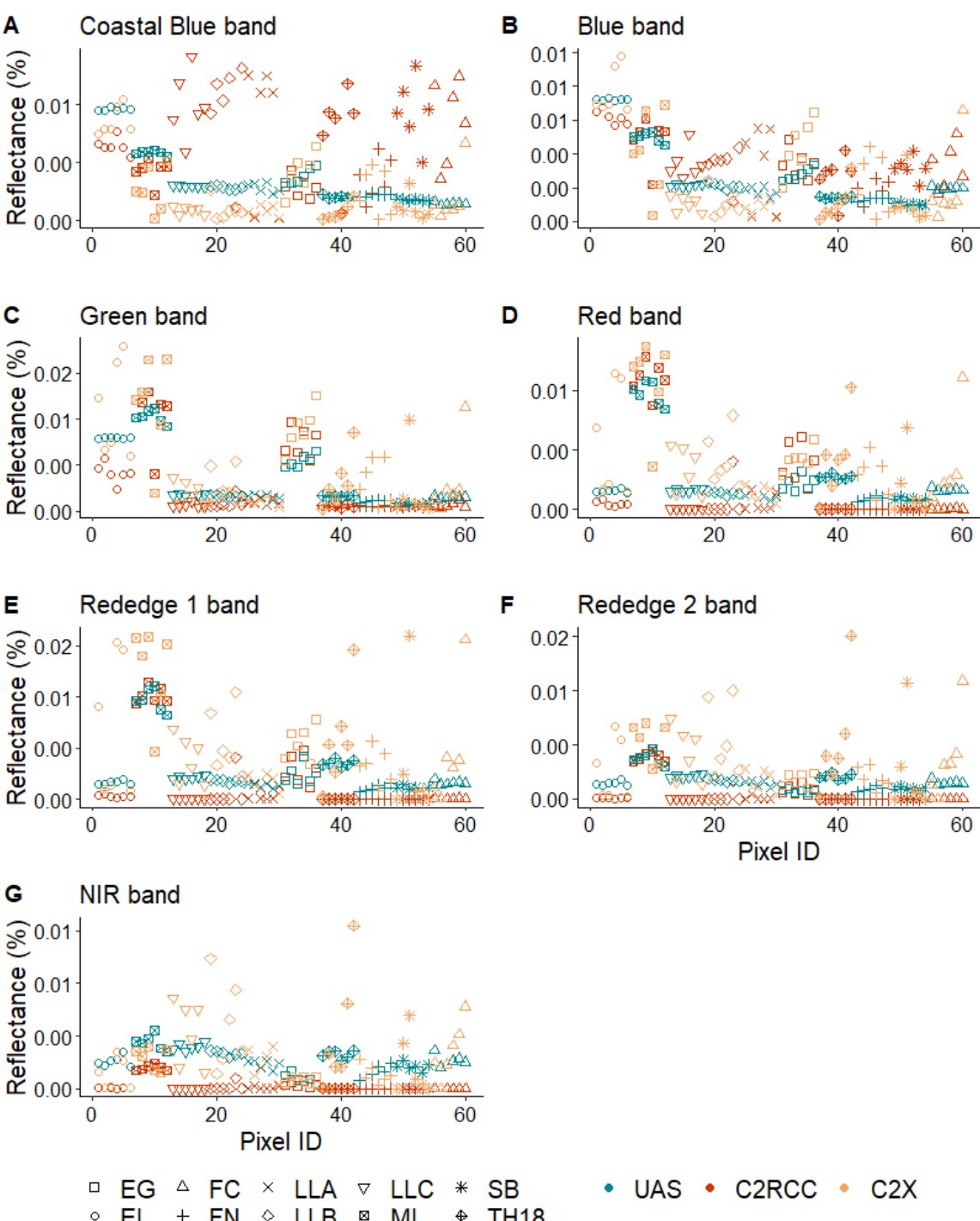

**Figure 4.** Pixel-by-pixel and band-by-band comparison of UAS remote sensing reflectance data with S2 atmospherically corrected products (CR2CC and C2X). The analyzed bands are costal blue (**A**), blue (**B**), green (**C**), red (**D**), rededge1 (**E**), rededge2 (**F**), and NIR (**G**). UAS data collected on 7 October and S2 data on 8 October 2020.

To assess the consistency of both sensors, Figures 5A and S2 shows the statistical analysis applied to all spectrum bands: $R^2$, RMSE, MAE, and bias, per band and per

AC. It can be observed that UAS vs. C2RCC performed better than UAS vs. C2X. The coefficients of determination obtained from comparing UAS orthomosaics and C2RCC were higher than those obtained with C2X for the green, red, red-edge1, red-edge2, and NIR bands (Figures 5A and S2). There is a weak relationship between the two sensors for both atmospheric corrections for the NIR band and there is an extremely weak relationship for the coastal blue band whose $R^2$ is the lowest of all bands. This may be because the coastal blue band usually has more residuals from the atmospheric correction than other visible bands. For the regression between UAS and C2RCC data, the results suggest that while more than 70% of the variation in the green, red and red-edge1 bands can be explained by the linear model, the remaining variation might be due to the differences such as acquisition dates, the spectral bands, bandwidth of the sensors, radiometric correction, and atmospheric residuals, among others [46,49].

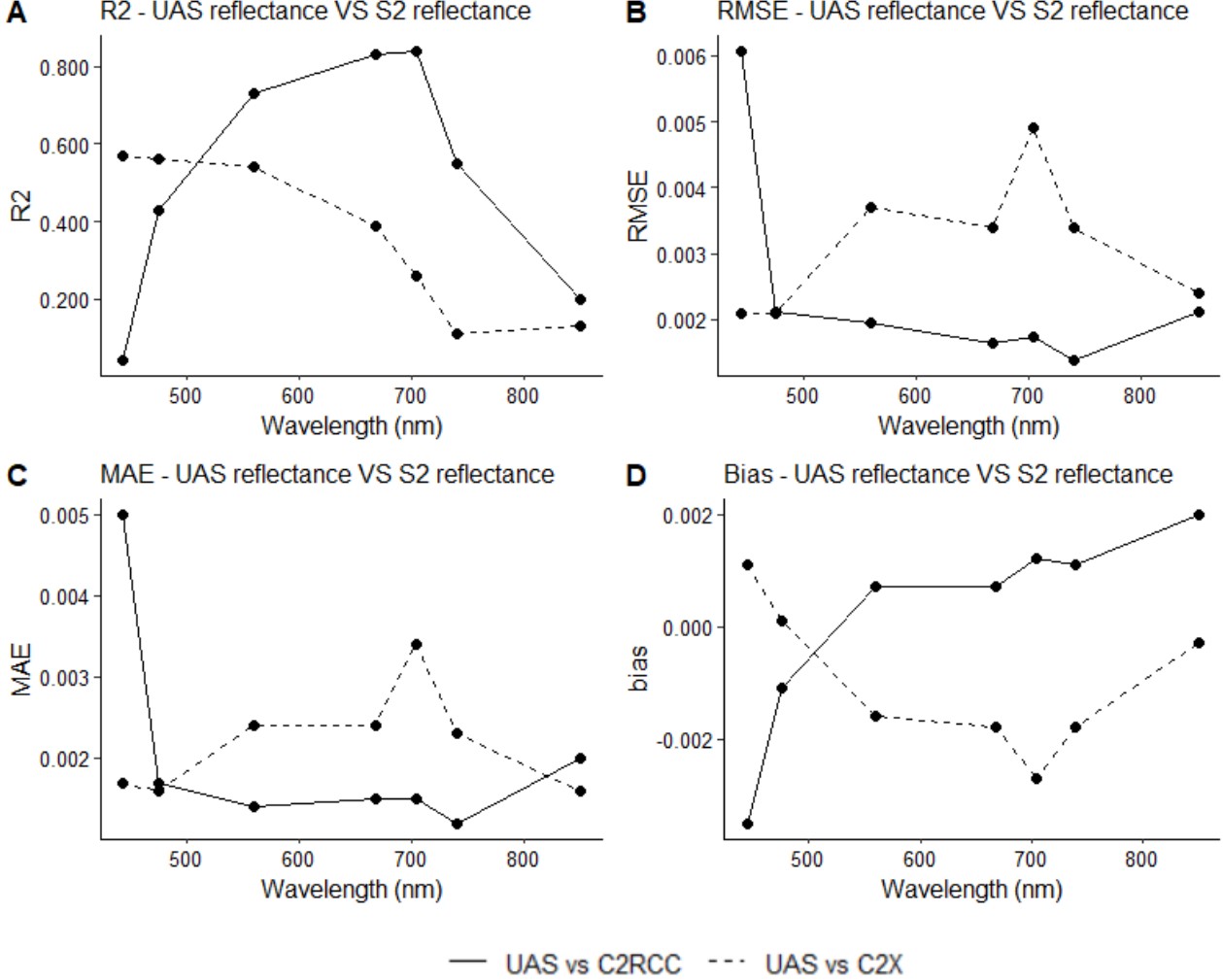

**Figure 5.** Band by band summary of statistics. (**A**) R2 (coefficient of determination); (**B**) RMSE (Root Mean Square Error) (1/sr), (**C**) MAE (Mean Absolute Error) (1/sr); (**D**) bias. UAS data of 7 October and S2 data of 8 October 2020.

Regarding the coastal blue band, the pixel-by-pixel comparison showed high variability between the UAS data and both satellite atmospheric corrections (Figure 4). Only in this case, C2X values were closer to the UAS data than C2RCC ($RMSE_{CB}$ = 0.0060 1/sr, $MAE_{CB}$ = 0.050 1/sr for C2RCC and $RMSE_{CB}$ = 0.0021 1/sr, $MAE_{CB}$ = 0.017 1/sr for C2X) (Figure 5B). The blue band has also high variability between the UAS data and both satellite atmospheric corrections and the RMSE values are the same ($RMSE_B$ = 0.0021 1/sr) for this

band while the MAE values are similar (MAE$_B$ = 0.017 1/sr for C2RCC and MAE$_B$ = 0.016 1/sr for C2X). The bias values of C2RCC show negative values for these bands, indicating an underestimation of the reflectance obtained by this AC processor, while C2X shows positive values indicating an overestimation of the reflectance values comparing to the UAS data.

Regarding the green, red, red-edge1, and red-edge2 bands, C2RCC values were generally closer to the UAS data (RMSE$_G$ = 0.0019 1/sr, RMSE$_R$ = 0.0016 1/sr, RMSE$_{RE1}$ = 0.0017 1/sr, RMSE$_{RE2}$ = 0.0014 1/sr) than the C2X (RMSE$_G$ = 0.0036 1/sr, RMSE$_R$ = 0.0034 1/sr, RMSE$_{RE1}$ = 0.0049 1/sr, RMSE$_{RE2}$ = 0.0034 1/sr) (Figure 5B). The MicaSense remote sensing reflectance data were always greater than the satellite's for the green, red, and red-edge1, except in the pixels corresponding to ML and EG (Figure 4C–E). C2RCC show a low positive bias, while C2X has negative values (Figure 5).

Regarding the NIR band, the pixel-by-pixel plot indicates that C2RCC and UAS have a more similar trend than C2X (Figure 4G), but both atmospheric corrections have similar RMSE values (RMSE$_{NIR}$ = 0.0021 1/sr for C2RCC, RMSE$_{NIR}$ = 0.0024 1/sr for C2X) in this spectral region. Furthermore, C2RCC increases the bias in the NIR band and C2X decreases it.

The Wilcoxon signed-rank test from the paired two samples showed a statistically significant difference between mean reflectances of UAS and mean reflectances of S2 based on a band-by-band comparison (*p*-value < 0.001) (Table 2). The *p*-value indicates strong evidence against the null hypothesis, as there is less than a 5% probability the null is correct. Therefore, the null hypothesis is rejected, and the alternative hypothesis is accepted. The only non-significant difference from the whole analysis was found in the blue (*p*-value = 0.214) and NIR band (*p*-value < 0.680) for the C2X processor.

**Table 2.** Wilcoxon signed-rank test of UAS and S2 remote sensing reflectance data.

| Bands | Coastal Blue | Blue | Green | Red | RedEdge1 | RedEdge2 | NIR |
|---|---|---|---|---|---|---|---|
| Observations | 60 | 60 | 60 | 60 | 60 | 60 | 60 |
| Spearman correlation | −0.243 | 0.546 | 0.685 | 0.338 | 0.277 | 0.000 | −0.084 |
| Wilcoxon signed-rank test for UAS and C2RCC data | <0.0001 | <0.0001 | 0.001 | 0.001 | <0.0001 | <0.0001 | <0.0001 |
| Wilcoxon signed-rank test for UAS and C2X data | <0.0001 | 0.214 | 0.002 | <0.0001 | <0.0001 | <0.0001 | 0.680 |

## 4. Discussion

The type of waters presented in this study corresponds to Case 2 waters, which have great variability in optical properties and can be classified as highly absorbing or highly scattering waters [38,50,51]. Dark reddish-brown acidic waters as can be seen in Figure 3C–E,G–J) are highly absorbing waters characterized by very low water-leaving reflectance and the maximum Rrs is typically <0.5%. While greenish-blue and greenish-brown waters, such as EL, ML, and EG, are highly scattering waters (Figure 3A,B,F).

The spatial resampling of the drone data from 8 cm to 10 m to match the satellite resolution provided reflectance values coherent with the values obtained from the S2 data, even though UAS data were more diverse in reflectance values than S2. The comparison of the UAS and S2 reflectance spectra for each water body revealed that C2RCC and C2X performed better for greenish-blue and greenish-brown water bodies concerning the UAS data, while for dark reddish-brown waters there is a lack of consistency in the form of the spectral signature obtained by both sensors (Figure 3). According to the band-by-band and pixel-by-pixel comparison and the statistics obtained, C2RCC data is more similar to the UAS data than C2X (Figure 4). However, despite the strong correlation found in the green, red, and red-edge1 bands for C2RCC, a significant difference in reflectance, between MicaSense RedEdge-MX Dual and S2 multispectral data, was found with the Wilcoxon signed-rank test (Table 2). This means that these differences in the reflectance values should

be considered when combining data from both sensors since these could have further consequences in developing multi-scale models.

The poor agreement between sensors may arise from different causes such as the spectral bands of the sensors, the bandwidth of the sensors, the radiometric correction, sensor degradation over time and the atmospheric conditions [46,49,52]. In this case, it is important to mention that the bandwidths of S2 were much wider than the UAS in several important bands, maybe due to spectral contamination. Radiometric resolution (number of data bits) is often overlooked and can be a factor along with signal to noise characteristics of the sensor. In this case, the signal-to-noise ratio (SNR), might be a problem since the Micasense was developed for land-studies, not water quality (requiring a higher SNR due in part to factors of water leaving radiances). In addition, it is well known that SNR in Sentinel 2 is low in blue-green bands. Hence, that difference in radiometric resolution can be a factor along with signal to noise characteristics of the sensors. One important factor is the temporal matching, there is one day difference between the UAS and S2 data. It is known that the dynamics of small water masses can greatly impact the results especially at very high spatial resolutions [53]. The dynamics of the water bodies may appear calm, but it is recommended to take matchups within a two-hour window around crossover time. Finally, even though the sensor MicaSense RedEdge is provided with DLS 2 and CRP to accomplish factory radiometric calibration, the calibration parameters may change when the camera is utilized in real-world conditions. As a result, previous studies recommend regular calibration of the radiometric calibration coefficients, proving that there is an impact of the radiometric correction on the spectral consistency between UAS and satellite data [52,54].

Other studies have found a relatively good correlation of bands between MicaSense RedEge sensor and S2. For instance, it has found that the spectral response of MicaSense RedEge sensor matches the response of S2 in a forest environment, except for the NIR band [55]. A good agreement in the shape of the spectral signature of a drinking water reservoir affected by eutrophication for both sensors has been showed, making it possible to develop a multi-scale monitoring tool [9]. On the other hand, the authors of [46] performed the t-test and Wilcoxon signed-rank test to compare the reflectance values on a crop environment from MicaSense RedEge sensor and S2 on a band-by-band basis obtaining the same result as ours, a significant difference in reflectances. It is important to mention that the analyzed water bodies with similar spectral signatures to the one studied by [9] (EG, EL, ML) have shown coherence between the shapes of the spectral curves of both sensors, but are the red AMD-affected water bodies the ones that did not show coherence in the results and should be further study to determine the cause of the difference in the sensors' responses. It is suggested that the spectrum of the AMD-affected water was not within the training database of the AC processors or the spectrum is out-of-scope of the algorithm definition, thus giving a water leaving reflectance spectrum with large deviations from the UAS data [40]. In this sense, new AC should be designed for this complex type of water. A recommendation to improve the obtained results is to verify the sensors data with field spectroradiometric measurements or hyperspectral sensor mounted on a UAS, especially in the dark reddish-brown acidic waters where UAS and S2 data failed to obtain similar reflectance data. Field spectroradiometric measurement of surface reflectance is a useful source of information that is completely unaffected by the atmosphere and, thus, is suitable for assessing the accuracy of the data obtained with other sensors [55,56].

## 5. Conclusions

This study dealt with a comparison between images of small water bodies derived from two different platforms: an UAS and the satellite S2. The UAS is an ultra-high-resolution platform and S2 a free medium resolution platform. The comparison was mainly based to determine whether the S2 sensor can complement the UAS sensor, in this case, the MicaSense RedEdge MX-Dual sensor, for inland water quality monitoring in mining environments. For the S2 data, two AC processors were applied to the imagery:

C2RCC and C2X, which have a good performance in complex waters and extreme complex waters respectively. Thus, the overall comparison was made between the UAS, C2RCC, and C2X datasets resampled to the spatial resolution of 10 m. Considering that the UAS data are a low altitude remote sensing tool with ultra-high spatial resolution (8 cm) that is not affected by the atmosphere and that field spectroradiometric measurements were not performed, the UAS data were considered as the ground truth reference source for the remote sensing reflectance. The results suggest that even though C2RCC showed a better performance than C2X compared to the UAS data, there is a significant difference in reflectance between MicaSense RedEdge-MX Dual and both AC processors which should be considered when combining data from both sensors since these could have further consequences in developing multi-scale models. The results obtained may be considered preliminary due to the limited in situ database and the lack of handheld spectroradiometric measurements to validate the performance of the UAS sensor and the AC processors.

**Supplementary Materials:** The following supporting information can be downloaded at: https://www.mdpi.com/article/10.3390/rs14164053/s1, Figure S1: Simplified orthomosaics of the surface reflectance building process: (a) images acquisition planning, (b) image alignment and 3D point cloud generation in Pix4Dmapper, and (c) reflectance map, in Filón Centro water body (Tharsis mine). Figure S2: Regression models between S2 (atmospherically corrected with C2X and C2RCC) and UAV-derived datasets for each analyzed band.

**Author Contributions:** Conceptualization, M.A.I., M.D.B and L.B.; methodology, M.A.I.; Sentinel-2 processing and correction, I.C.; UAS processing, M.A.I., M.D.B. and L.B.; validation, M.A.I.; writing—review and editing M.A.I., M.D.B., I.C. and L.B. All authors have read and agreed to the published version of the manuscript.

**Funding:** This study was supported in part by the Erasmus Mundus Joint Master Degree (EMJMD) in Water and Coastal Management (WACOMA) with the contribution of the Erasmus+ Programme of the European Union. This work was also supported by Plan Andaluz de Investigación RNM 166 Environmental radioactivity research group (LB) and partially by the FEDER UHU2020-21 Project. UAS equipment from University of Cádiz Drone Service was supported by MINECO infrastructure projects (EQC2018-00446-P and UNCA-2013-1969). M.D.B. thanks the Spanish Ministry of Science and Innovation for the Postdoctoral Fellowship granted under application reference IJC2018-035056-I. This research was also funded by the Spanish Ministry of Science and Innovation, the Spanish State Research Agency, the European Regional Development Fund MCIN/AEI/10.13039/501100011033 (Sen2Coast Project; RTI 2018-098784-J-I00), and the Consejería de Transformación Económica, Industria, Conocimiento y Universidades from Andalusian Government through the Andalusian FEDER operational program 2014–2020 (A1123060E0_PYC20 RE 032 UHU) and the call 2020 for collaborative interest projects in the field of the Innovation Ecosystems of the International Excellence Centers.

**Data Availability Statement:** Not applicable.

**Acknowledgments:** We thank to Tharsis Mining & Metallurgy for allowing us to collect the samples in the abandoned mining areas that belong to the company since 2018. The authors also gratefully acknowledge the support of the Guest Editor, the Assistant Editor and the three anonymous reviewers for their comments and positive criticisms, which notably improved the quality of the original paper.

**Conflicts of Interest:** The authors declare no conflict of interest.

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
