# Peer review of "Comparison of UAS and Sentinel-2 Multispectral Imagery for Water Quality Monitoring: A Case Study for Acid Mine Drainage Affected Areas (SW Spain)"

_remotesensing, doi:10.3390/rs14164053_

Round 1

Reviewer 1 Report

Review for Manuscript remotesensing-1797416

This is an interesting work based on comparison of state-of-the-art drone multispectral imagery and medium resolution Sentinel-2 imagery for environmental management. The authors promote the usability of multispectral data for environmental monitoring of acidic lakes adding a new application in the growing field of drone-based remote sensing. This manuscript is well structured and the findings of this study are clearly presented and discussed. Thus I recommend a minor revision and below I have comments for a few additions that will help finalizing this manuscript.

Introduction Lines 58-70

Although the authors address the environmental aspect of their study in this paragraph, I would recommend providing some additional information about the impact of AMD on spectral characteristics of the lakes. This will help with understanding the coupling between chemical and optical properties of water and how remote sensing is applied for resolving these (lines 307-311 should be moved here). Please also mention any previous remote-sensing studies on this topic and the novelty of your application.

Methodology Lines 182-191

 Please add a few sentences with details about how each AC algorithm works. E.g.: Which bands are used and what other parameters are taken into consideration. (lines 320-324 should be moved here).

Discussion

It would be good to discuss (a small paragraph) about whether drone imagery requires some AC or additional radiometric processing for compensating local effects on water reflectance.

This article may help: https://doi.org/10.1016/j.rse.2021.112691

Please consider adding a few sentences with the characteristics of the ideal AC algorithm regarding the study of acidic lakes. Please mention any other software tools that should be tested in the future (e.g.: WASI?).

Reviewer 2 Report

The integration of multi-platform remote sensing approaches to assist in the interpretation of environmental processes and to improve the accuracy of a range of atmospheric correction and analysis algorithms is a very valuable and appropriate scientific approach. In order for different sensors to be integrated accurately all potential sources of error have to be accounted for and hopefully fully addressed. 

The use of remote sensing for the the monitoring of the environmental impacts of mining is a very valid and useful application of remote sensing methodologies.

Reviewer 3 Report

A lot of good work was reflected in this manuscript.  I have a few edit suggestions and have attached a Word doc file as well.

You may want to consider having you main objective (lines 70-74) additionally in the abstract, as it is very clear and important for environmental monitoring.

Line 154 suggested edit for clarity such as, ...and discarded "if blurred". rather than ...and discarded "when being blurring".

Please define QGIS in line 197

Please define the symbol Pi used in line 202, for those readers who are not familiar with RRS, so that they don't need to read references [32,33].  It is a standard algorithm, but you will probably have a diverse audience of readers.

The authors conducted field work taking in situ measurements as well as samples later analyzed at the Institute of Environmental Assessment and Water Research (lines 105-115).  They may want to say something regarding this work or perhaps that these data will be published separately.

Line 286 suggest "greater" rather than "above"

Round 2

Reviewer 2 Report

All remote sensing data acquired from a platform must be atmospherically corrected. When data is collected on different dates it must be corrected using a systematic, accurate methodology or the results are meaningless.

A clearer understanding of what information on water quality the remote sensing signal is giving is required.